# Progress and Challenges to Hepatitis E Vaccine Development and Deployment

**DOI:** 10.3390/vaccines12070719

**Published:** 2024-06-28

**Authors:** Xingcheng Huang, Jiaoxi Lu, Mengjun Liao, Yue Huang, Ting Wu, Ningshao Xia

**Affiliations:** 1State Key Laboratory of Vaccines for Infectious Diseases, Xiang An Biomedicine Laboratory, School of Public Health, Xiamen University, Xiamen 361000, China; 32620221150913@stu.xmu.edu.cn (X.H.); jiaoxi@stu.xmu.edu.cn (J.L.); liaomengjun@stu.xmu.edu.cn (M.L.); huangyuesph@xmu.edu.cn (Y.H.); 2State Key Laboratory of Molecular Vaccinology and Molecular Diagnostics, National Institute of Diagnostics and Vaccine Development in Infectious Diseases, National Innovation Platform for Industry-Education Integration in Vaccine Research, The Research Unit of Frontier Technology of Structural Vaccinology of Chinese Academy of Medical Sciences, Xiamen University, Xiamen 361000, China

**Keywords:** hepatitis E, HEV, vaccine, clinical trial, outbreaks, progress

## Abstract

Hepatitis E is a significant cause of acute hepatitis, contributing to high morbidity and mortality rates, and capable of causing large epidemics through fecal–oral transmission. Currently, no specific treatment for hepatitis E has been approved. Given the notably high mortality rate among HEV-infected pregnant women and individuals with underlying chronic liver disease, concerted efforts have been made to develop effective vaccines. The only licensed hepatitis E vaccine worldwide, the HEV 239 (Hecolin) vaccine, has been demonstrated to be safe and efficacious in Phase III clinical trials, in which the efficacy of three doses of HEV 239 remained at 86.6% (95% confidence interval (CI): 73.0–94.1) at the end of 10 years follow-up. In this review, the progress and challenges for hepatitis E vaccines are summarized.

## 1. Introduction

Hepatitis E virus (HEV) is a major infective cause of acute hepatitis, causing an estimated 3.3 million cases annually [1]. As members of the genus *Paslahepevirus*, the HEV strains commonly implicated in human infections are HEV-1, HEV-2, HEV-3, and HEV-4, all of which belong to the subfamily *Orthohepevirinae* [2]. HEV-7 and HEV-8, first detected in camels, have been demonstrated to cross-transmit to non-human primates, with one report of HEV-7 infection in humans [3,4]. Recently, a rat hepatitis E virus (*Rocahepevirus ratti*) has been recognized as an emerging genotype, which may establish persistent infections in immunocompromised individuals or even immunocompetent individuals [5]. An epidemiological study in Hong Kong found that HEV-C1 infections constitute 8% of all genotyped hepatitis E cases [6]. 

Serological studies have estimated that one-third of the global population has been infected with HEV [7]. HEV-1 and HEV-2 primarily infect humans, with outbreaks frequently occurring in hyperendemic regions, such as Asia, Africa, and the Middle East. HEV-1 and HEV-2 outbreaks are often catalyzed by the accidental contamination of water sources with feces following heavy rainfall or flooding [8]. On the contrary, HEV-3 and HEV-4, primarily found in developed countries, are associated with foodborne infection or direct contact with animal reservoirs, specifically domestic pigs and wild boars [9]. More specifically, prevalent HEV genotypes have shifted following local economic development. In China, improvements in community hygiene since 2000 have led to a shift in the predominant genotype from HEV-1 to HEV-4 [10]. 

Although typically resulting in self-limiting acute viral hepatitis, HEV-related infections pose a significant public health challenge, particularly in low-income countries. Both symptomatic infection and seroprevalence rates increase significantly with age [11,12]. Acute HEV infection in pregnant women may result in fulminant hepatic failure, an explosive disease with a mortality rate of up to 30% [13]. Additionally, vertical transmission from the mother to the child during gestation is clear [14,15,16]. Of note, individuals who are immunocompromised or have undergone solid organ transplants may have difficulty in clearing hepatitis E virus if infected with HEV-3 or HEV-4. This susceptibility can lead to progressive liver fibrosis and even liver failure [17,18]. In patients with pre-existing chronic liver disease (CLD), HEV infection is a potential trigger of acute-on-chronic liver failure, which accelerates disease progression and increases the mortality rates in these patients [19,20]. 

To date, specific drugs approved for the treatment of HEV infection are limited [21,22]. The primary strategy against hepatitis E focuses on prevention. The development of inactivated or live attenuated virus vaccines has been challenged by the inefficient replication of HEV in cell cultures [23], so the recombinant HEV vaccines were considered a promising strategy. The only commercialized HEV vaccine is a recombinant vaccine, HEV 239 (Hecolin, Xiamen Innovax Biotech, Xiamen, China), which was licensed in China in 2012 and, subsequently, in Pakistan in 2020. 

## 2. Clinical Progress

HEV has single-stranded RNA consisting of three overlapping open reading frames (ORF1–3) in its 7.5 kb genome. Recently, a novel ORF-4 has been identified solely in HEV-1 [24]. The capsid protein encoded by ORF2, consisting of 660 amino acids, is crucial for binding to host cells and eliciting neutralizing antibodies. Consequently, it has been ranked as a candidate target site for recombinant HEV vaccine development [25]. Currently, four recombinant hepatitis E vaccines have reached the clinical stage (Table 1). The first candidate vaccine, rHEV, was a 56-kDa baculovirus-expressed virus-like particle (VLP) developed by GlaxoSmithKline (GSK, Brentford, United Kingdom) and the National Institutes of Health (NIH, Bethesda, MD, USA). The safety and efficacy of rHEV were successfully evaluated in Phase II trials in Nepal from 2000 to 2004 [26]. Unfortunately, no further progress has been reported on the development of this vaccine. Another hepatitis E vaccine, HEV 239, is a 26-kDa recombinant polypeptide that corresponds to 368–606 aa of the capsid protein from a HEV-1 strain. Being licensed for use in individuals aged >16 years, HEV 239 is available as a three-dose schedule (0, 1, and 6 months), with each dose containing 30 μg of the antigen. More recently, HEV P179 (Changchun institute of Biological Products Co., Ltd, Changchun, China) has been developed based on the 439–617 aa of capsid protein from a HEV-4 strain. The safety of HEV P179 has been evaluated in a Phase I clinical trial in Jiangsu Province, China [27]. The fourth recombinant hepatitis E vaccine, developed by Zydus Lifesciences Ltd. (Ahmedabad, India), is currently undergoing a Phase II clinical trial in India [28].

## 3. Hepatitis E Vaccine Efficacy and Safety in Clinical Trials

Both rHEV and HEV 239 vaccines have demonstrated efficacy and safety in published clinical studies (Table 2). In the Phase I clinical trials of rHEV, 88 healthy volunteers aged 18–50 years received three doses at 0, 1, and 6 months, and 22 volunteers were administered doses at 1, 5, 20, and 40 μg for dose escalation. All of the formulations showed high immunogenicity with good tolerability, with the 20 μg formulations being selected for subsequent development [12]. In a double-blind, placebo-controlled Phase Ⅱ trial, 2000 anti-HEV-negative healthy male adults were randomly assigned to receive three doses of 20 μg of the rHEV vaccine or placebo. During the 2-year follow-up period post-vaccination, hepatitis E occurred in 66 of the 896 individuals (7.4%) in the placebo group compared to 3 of the 898 individuals (0.3%) in the vaccine group, indicating a vaccine efficacy of 95.5% (95% confidence interval (CI), 85.6–98.6) [26]. Despite these promising outcomes, GSK unexpectedly discontinued the development of the rHEV vaccine [32]. 

The HEV 239 vaccine, co-developed by Xiamen Innovax Biotech and Xiamen University, is the most extensively studied vaccine, and has been demonstrated to be safe and effective in individuals >16 years of age. HEV 239 has been comprehensively evaluated through Phase I to Phase IV clinical trials in China. Additionally, in October 2017, a Phase IV clinical trial was conducted to evaluate its effectiveness and safety among childbearing women in Bangladesh [33]. Another randomized Phase I clinical trial was conducted in the U.S. in 25 participants aged 18 to 45 years to evaluate the safety and immunogenicity [34]. For emergency use, the first vaccination campaign using Hecolin was conducted in 2022 during an outbreak in an internally displaced persons (IDP) camp in South Sudan, covering more than 40,000 people, including pregnant women [35]. 

In the Phase I clinical trial, 44 healthy volunteers aged 21–55 years were enrolled to receive 20 μg of the HEV 239 vaccine at month 0 and 1, with a follow-up extending to 60 days after the initial dose. The results showed that the safety of the vaccine was good, and no abnormal changes were observed in the blood biochemical markers. No serious adverse events (SAE) related to the vaccine were reported during the follow-up period [36].

The Phase II clinical trial was separated as Phase IIa and Phase IIb, with an 18-month follow-up period [37]. In the dose-scheduling component, 457 healthy adults were randomly assigned to three groups, which were vaccinated with two or three doses of 20 μg of the HEV 239 vaccine at 0, 1, and 6 months or 0 and 1 months, respectively, and three doses of the commercial hepatitis B vaccine at 0, 1, and 6 months as a control. In the dose-escalation component, 155 healthy adults were randomized in four groups, with each group receiving the HEV 239 vaccine at 0, 1, and 6 months in doses of 10, 20, 30, and 40 μg. The vaccine was well tolerated, and no statistically significant difference in the localized adverse reactions between the two-dose and three-dose regimens were reported (*p* > 0.05). The rates of systemic adverse reactions and SAE were comparable to those of the control group (*p* > 0.05). Both the three-dose group and the two-dose group showed a high rate of anti-HEV IgG seroconversion (100% vs. 98%). Notably, a higher geometric mean concentration (GMC) of anti-HEV IgG was observed in the three-dose group compared to the two-dose group (15.9 vs. 8.6 WHO units (Wu)/mL, *p* < 0.05) [37]. The dose escalation component revealed the IgG titers increased progressively with the increasing dosage from 10 to 40 μg. The adverse event rates remained consistent across the four dosage groups, indicating the vaccine was well tolerated and immunogenic at all dosage levels. Additionally, the incidence of subclinical HEV infection (indicated by spontaneous seroconversion or a >3-fold increase in anti-HEV IgG level) was significantly reduced following the second and third doses of the vaccine, suggesting its protection against new infections. After completing the Phase II clinical trial, a 0-, 1-, and 6-month schedule using a 30 μg formulation was selected for the Phase III clinical trial.

In 2007–2009, a randomized, double-blind, controlled Phase III clinical trial of the HEV 239 vaccine was conducted in 11 rural townships in Dongtai city in Eastern China. To evaluate the efficacy of the vaccine, a symptom-based active hepatitis surveillance system including 205 sentinels was established, which comprised virtually all of the village clinics, the township hospitals, and public and private clinics, covering all of the residents in the 11 rural townships [29].

A total of 112,604 healthy subjects aged 16 to 65 years old were enrolled, stratified by age and sex, and randomly assigned to the vaccine and control groups. The vaccine group was administered 30 μg of HEV 239, while the control group received 5 μg of the hepatitis B vaccine, both following to a 0-, 1-, and 6-month schedule. The results indicated that 52% of the subjects were anti-HEV IgG negative at the baseline, and 99.9% of the negative subjects seroconverted after receiving three doses of the HEV 239 vaccine. Most of the adverse reactions were mild, with the incidence of serious adverse events (SAEs) being similar in the vaccine and control groups (*p* > 0.05). No SAEs were related to vaccination. The trial was successful in achieving its primary aim, which was to prevent symptomatic HEV infection. The vaccine showed a 100% efficacy rate (95% CI: 72.1–100) in those who received three doses of the HEV 239 vaccine according to the protocol, which included hepatitis E cases occurring during 12 months from the 31st day after the third dose. Overall, 23 cases of hepatitis E occurred between the first immunization and month 19, including 1 in the vaccine group and 22 in the control group, resulting in a protection rate of 95.5% (95% CI: 66.3–99.4). Among these cases, 13 cases were successfully genotyped, revealing 12 with HEV-4 and 1 with HEV-1, highlighting the cross-protective efficacy of the HEV 239 vaccine.

The initial study ended on month 19. All of the participants remained blinded, and the follow-up extended to month 55. During this extended period, 37 new cases were confirmed. Among the participants who received three doses, a total of 48 cases occurred from month 7 to month 55, including 3 cases in the vaccine group and 45 cases in the control group, demonstrating a protective efficacy of 93.3% (95% CI: 78.6–97.9). For those who received at least one dose, 60 cases occurred from the first dose through to month 55, with 7 cases in the vaccine group and 53 cases in the placebo group, resulting in an efficacy rate of 86.8% (95% CI: 71.0–91.9) [30]. 

Based on these findings, a 10-year follow-up study of HEV 239 was conducted to assess the long-term efficacy [31]. In addition to the participants from the Phase III clinical trial, an additional 178,236 residents of the study region, who were within the age range of the study participants but did not participate in the Phase III trial, were included as an external control group. Over the 10-year period, 415 cases of hepatitis E were identified, including 13 cases (0.2 per 10,000 person-years) in the vaccine group, 77 cases (1.4 per 10,000 person-years) in the placebo group, and 325 cases (1.8 per 10,000 person-years) in the external control group. Among the participants who received at least one dose and were followed from the onset of the initial study, the 10-year efficacy rate was 83.1% (95% CI: 69.4–91.4) when compared to the placebo group and 88.0% (95% CI: 79.1–93.7) in comparison with the external control group. Furthermore, 70% of the subjects who received three dose and were seronegative at the baseline maintained detectable levels of anti-HEV IgG antibodies in the 8.5-year period. The success of these clinical trials for the HEV 239 vaccine has stimulated enthusiasm and optimism in the campaign against hepatitis E.

More recently, another E. coli-expressed hepatitis E vaccine, p179, has demonstrated its safety in Phase I clinical trials [27]. This study was designed to consist of a dose escalation scheduling of 20 μg, 30 μg, and 40 μg P179 components, with 30 μg of HEV 239 as a control. A total of 120 eligible participants aged 16–65 years were randomized and vaccinated at 0, 1, and 6 months. All three of the different dosages of the HEV p179 vaccine showed safety and good tolerance. Similar solicited total and systemic adverse reaction incidence of the P179 groups and the control group were observed (*p* > 0.05), as well as the changes in the blood routine and serum biochemical indexes [27]. A Phase II clinical trial of P179 is ongoing. For Lipo-NE-P, developed by Zydus Lifesciences Ltd., no results have been published on Clinical-Trials.gov or in the academic literature.

## 4. Hepatitis E Vaccination with Two-Dose Schedule and Accelerated Schedule

In hepatitis E epidemic settings, such as IDP camps in South Sudan, where formalized infrastructure and coordinated humanitarian response are not effective in controlling transmission, vaccination against hepatitis E could be important in confining the outbreak [35]. The routine hepatitis E vaccination schedule comprises doses at 0, 1, and 6 months, however, the frequent movements of people for better living conditions in the IDP camps make it challenging to provide timely protection for them. Thus, shorter immunization schedules will be necessary. In the Phase Ⅲ study of HEV 239, 15 days after the second dose and before the third dose, five cases occurred in the control group, while there were no cases in the vaccine group, resulting a 100% (95% CI: 9.1–100) protection rate in 4.5 months with the two-dose regimen [30]. Up to month 120 in the extension study, the two-dose regimen demonstrated an 89.9% (95% CI: 43.4–99.7) efficacy rate vs. the external control group [31]. Among the participants who were seronegative before vaccination, the two-dose regimen with a one-month interval showed comparable seropositive rates (100% vs. 100%) at month 7 (6 months after the second dose) compared with those who received the three-dose at month 13 (7 months after the third dose), but a lower GMC of anti-HEV IgG [2.06 Wu/mL (95% CI: 1.68–2.51) vs. 4.09 Wu/mL (95% CI: 3.81–4.39)] [30]. Similar results were observed in a Phase I study in U.S. seronegative adults, in which the GMCs at 28 days after dose 2 and dose 3 were 6.16 Wu/mL (95% CI: 4.40–8.63) and 11.50 Wu/mL (95% CI: 7.90–16.75), respectively [34]. Additionally, a randomized, controlled trial conducted in Bangladesh evaluated the immunogenicity and safety of a two-dose schedule of HEV 239 administered at a one-month interval. In this study, 100 individuals aged 16–39 years were randomized to receive either two doses of the HEV 239 or the HBV vaccine, following a 0- and 1-month schedule, with follow-up extending to 23 months after the last dose. All of the seronegative participants in the HEV 239 group seroconverted at month 2, maintained a 98% positivity rate at 24 months, and exhibited elevated HEV IgG antibody levels compared to the HBV vaccine group (*p* < 0.001), with geometric mean titers of 20.2 vs. 0.09 Wu/mL at month 2 and 1.6 vs. 0.11 Wu/mL at month 24 [38]. 

In addition to the two-dose data, an accelerated three-dose schedule was also evaluated. Z. Chen et al. conducted a Phase IV trial to assess the safety and immunogenicity of an accelerated HEV 239 schedule at 0, 7, and 21 days. The study randomized 126 anti-HEV seronegative adults to receive either the accelerated schedule or the routine schedule. The results indicated that all of the participants in both the accelerated and the routine groups were seropositive at 1 month after the three doses (57/57 and 63/63, respectively), with no significant differences in the GMC of anti-HEV IgG being noted between the two groups (8.51 vs. 9.67 Wu/mL). The incidence rates of solicited adverse reactions were comparable (32.26% vs. 30.16%, *p* = 0.800). These findings suggest that the immunogenicity of the accelerated 0-, 7- and 21-day schedule in adults is non-inferior to that of the routine regimen in adults [39].

## 5. Hepatitis E Vaccine in Elderly People

Epidemiological studies have identified that the highest incidence and mortality rates of hepatitis E occur in adults over the age of 60. To evaluate the safety and immunogenicity of HEV 239 in the elderly population (aged > 65 years), an open-label, controlled study was conducted [40]. A total of 200 elderly people aged > 65 years and 201 adults aged 18–65 years received HEV 239, according to the routine schedules. At month 7, 96.7% of the elderly people and 98.9% of the adults aged 18–65 years seroconverted. The GMCs of anti-HEV IgG were 5.36 Wu/mL (95% CI: 3.88–7.41) and 10.84 Wu/mL (95% CI: 9.42–12.47) in the baseline seronegative elderly people and the adults aged 18–65 years, respectively. The HEV 239 was well tolerated among the elderly, with 40% reporting adverse events, which is comparable to the 42.3% reported in the 18–65 age group. No vaccine-related SAEs were reported.

## 6. Hepatitis E Vaccine in Pregnant Women

HEV infection poses a high maternal morbidity and mortality for pregnant women during hepatitis E outbreaks. Previous studies on HEV vaccines and maternal outcomes are still limited. In the Phase Ⅲ trial of HEV 239, 37 subjects in the vaccine group and 31 subjects in the placebo group became pregnant and were inadvertently vaccinated. Wu et al. [41] compared the pregnancies in the HEV 239 and placebo groups with matched non-pregnant women for adverse events and pregnancy outcomes in a preliminary post hoc analysis. Only one pregnant person receiving HEV 239 reported mild local pain, and the rate of adverse events for both the HEV 239 and the placebo groups were similar to those of the matched non-pregnant women. Elective abortions were reported by 51.3% in the vaccine group and 45.2% in the placebo group, and the remaining pregnancies were delivered vaginally or via C-section, with no spontaneous abortions or infant malformations. Subsequently, a double-blinded, cluster-randomized Phase IV trial was conducted in Bangladesh in 2017, in which about 20,000 women aged 16–39 years were randomly allocated in a 1:1 ratio to receive either the HEV 239 or the hepatitis B vaccine. The primary outcome was a confirmed HEV infection in the pregnant women; moreover, the safety and immunogenicity of the vaccine were evaluated as well. The participants who became pregnant during the 2-year follow-up period were visited every 2 weeks to collect the pregnancy outcomes and were monitored for clinical hepatitis [33]. The study finished in 2022, but the findings have not yet been disclosed. 

In 2023, Guohua et al. [42] conducted a safety post hoc analysis on data from the Phase III clinical trial of human papillomavirus (HPV) type 16/18 vaccine (Cecolin), which provided more robust safety evidence. A total 7372 healthy women aged 18–45 years old were randomly assigned to receive three doses of the control vaccine Hecolin (*n* = 3683) or the test vaccine Cecolin (*n* = 3689) at 0, 1, and 6 months. Finally, 140 pregnant women experienced 143 pregnancy events following the vaccination. The incidences of adverse reactions were 31.8% in Hecolin recipients compared to 35.1% in those receiving Cecolin (*p* = 0.6782). In the analysis of the impact of vaccine exposure for pregnancy on adverse pregnancy outcomes and pregnancy complications, proximal exposure was defined as vaccination during pregnancy or the onset of pregnancy within 90 days post any dose, and pregnancy beyond 90 days after vaccination was distal exposure. Compared with HPV vaccination, no significantly increased risk of abnormal fetal loss (OR = 0.80, 95% CI: 0.38–1.70) or neonatal abnormalities (OR = 2.46, 95% CI: 0.74–8.18) were observed in the women with proximal exposure to HEV vaccination, as did distal exposure. This post hoc analysis suggested no increased risk from the HEV vaccination in pregnant women compared to the HPV vaccination [42].

Although the current evidence does not support the use of hepatitis E vaccination in routine immunization for pregnant women, the World Health Organization (WHO) issued statements in 2021 to recommend the vaccination of pregnant women against hepatitis E during outbreaks [43]. In 2022, a vaccination campaign against hepatitis E was implemented in the Bentiu internally displaced persons camp in South Sudan [35]. Approximately 40,000 (86%) residents of the Bentiu IDP camp aged 16–40 years, including pregnant women, received at least one dose of the HEV 239 vaccine. In the shelter survey, 91 (7.6%) of the 1195 individuals reported new symptoms within 72 h following the vaccination. Of these, women experienced new symptoms more frequently than men (73 (10%) vs. 18 (3.7%), *p* < 0.001). Among 118 pregnant women identified with a known HEV 239 vaccination, 1 reported abdominal cramps, which resolved two hours following the onset of the symptoms. To establish the safety and immunogenicity of Hecolin during pregnancy, the International Vaccine Institute (IVI) plans to initiate a Phase II, randomized, observer-blind, controlled clinical trial to evaluate the safety and immunogenicity of HEV 239 in pregnant women in Pakistan in 2024 [44].

## 7. Hepatitis E Vaccine in Patients with CLD

Hepatitis E infection poses a significant risk to individuals with pre-existing liver diseases, usually triggering acute-on-chronic liver failure and increasing mortality rates among these patients [20,21]. In an extended analysis of the HEV 239 Phase III clinical trial, 406 subjects in the vaccine group and 424 in the placebo group were hepatitis B surface antigen (HBsAg)-positive. After a routine vaccination schedule, both HBsAg-negative and HBsAg-positive individuals exhibited comparable levels of anti-HEV-IgG antibodies (19.32 vs. 19.00 Wu/mL). The solicited adverse event rates were similar between HBsAg-negative and HBsAg-positive individuals in the vaccine group (13.61% vs. 11.58%, *p* > 0.05) [45]. Furthermore, a subsequent Phase Ⅳ clinical study was conducted to evaluate the immunogenicity and safety of HEV 239 in chronic hepatitis B (CHB) patients in China [46]. The findings confirmed that the hepatitis E vaccine was safe and well tolerated among CHB patients, with no significant clinical changes observed in the liver function indicators. At month 1 after the routine schedule, the seroconversion rates were >97% in both the CHB and the control groups. Additionally, the GMC of anti-HEV IgG was non-inferior in the CHB group, with a ratio of 69% (95% CI: 55–85) compared to the control group.

**Table 2 vaccines-12-00719-t002:** Clinical studies of the HEV 239 vaccine.

NCT Number	Study Start	Primary Objectives	Enrollment	Vaccination Schedule	Phases	Country	Status	Ref
ACTRN12607000368437	January 2005	Safety and immunogenicity, exploration of dose and schedule	457 for dose-scheduling; 155 for dose escalation	0, 1, 6 m or 0, 6 m	II	China	Completed	[37]
NCT01014845	August 2007	Efficacy and safety in healthy adults	112,604, 16–65 y	0, 1, 6 m	III	China	Completed	[29,30,31]
NCT02417597	April 2015	Efficacy and safety in elderly people >65 years	400 for >65 y; 201 for 18–65 y	0, 1, 6 m	IV	China	Completed	[40]
NCT02584543	October 2015	Safety and immunogenicity of coadministration with the HBV vaccine	602, ≥18 y	0, 1, 6 m	IV	China	Completed	[47]
NCT02964910	August 2016	Efficacy and safety in people with chronic HBV	475, ≥30 y	0, 1, 6 m	IV	China	Completed	[46]
NCT03168412	May 2017	Immunogenicity and safety of an accelerated vaccination schedule	126, ≥18 y	0, 7, 21 d or 0, 1, 6 m	IV	China	Completed	[39]
NCT02759991	October 2017	Immunogenicity and safety of a two-dose schedule	100, 16–39 y	0, 1 m	II	Bangladesh	Completed	[38]
NCT02759991	October 2017	Effectiveness and safety in women of childbearing age in Bangladesh	19,460, 16–39 y	0, 1, 6 m	IV	Bangladesh	Completed	-
NCT03827395	April 2019	Safety and immunogenicity in U.S. adults	25, 18–45 y	0, 1, 6 m	I	USA	Completed	[34]
NCT06306196	April 2024	Immunogenicity and safety of Hecolin in HIV-positive/negative adults and in children	410, 18–45 y; 175, 12–17 y; 175, 6–11 y; 100, 2–5 y	0, 1, 6 m	IIb	South Africa	Active, not recruiting	-
NCT05808166	June 2024	Safety and immunogenicity in pregnant Pakistani women	2358, 16–45 y	0, 1, 6 m	II	Pakistan	Active, not recruiting	-

## 8. Challenges in Hepatitis E Vaccine Development

Hepatitis E is widely recognized as the most common cause of acute viral hepatitis globally. During 2010–2020, 12 independent outbreaks occurred in different countries, resulting in more than 30,000 symptomatic cases [48]. A promising hepatitis E vaccine has been first licensed in China and was recommended to prevent outbreaks of hepatitis E and mitigate the consequences for high-risk populations in the current WHO position paper [43]. However, the absence of WHO pre-qualification and limited immediate availability of Hecolin complicate its deployment for both political and practical reasons. To facilitate access to the vaccine in low-income countries, the Global Alliance for Vaccines and Immunization (GAVI) has included the HEV vaccine in its 2024 investment strategy [49].

Despite the evidence of heat stability being 2 months at 30–37 °C, the package insert of Hecolin recommends cold-chain storage (2–8 °C). Prefilled syringes provide precise dosing and lower the risk of contamination, but cause more complexities in transportation, storage, and waste management. Furthermore, for mass vaccination efforts in developing countries, a multi-dose packaging option would be preferable, while requiring process changes and further stability studies. A shorter vaccination schedule is desirable in outbreak control situations. The available data have suggested a rapid and strong anti-HEV antibody response and high protection following a two-dose regimen, which supports its implementation during outbreaks. However, more robust evidence is needed in order to confirm its long-term protection. 

Hecolin is licensed for individuals who are older than 16 years of age. Nonetheless, epidemiology studies have reported that more than 18% prevalence of anti-HEV IgG was found in children under 15 years of age in developing countries [50]. A survey conducted in the Bentiu refugee camp highlighted that, among the 670 suspected cases in the initial quarter of 2022, 422 (63%) suspected cases were reported in persons younger than 16 years of age, suggesting the urgent demand for vaccination in those individuals [51]. There are notable gaps and deficiencies in safety and protective data among high-risk populations, including pregnant women, individuals with chronic liver disease, and the immunosuppressed. Although long-term protective effects have been demonstrated by a ten-year follow-up study, whether a booster dose is necessary remains unclear.

Cross-reactivity among different HEV genotypes infecting humans has been demonstrated in vitro. Therefore, the HEV vaccine is expected to provide cross-genotype protection against all four genotypes. During the Phase III clinical trials of the HEV 239 vaccine, sufficient data supported the cross-protection of HEV 239 against HEV-4, but not for HEV-1 or HEV-2. The current research in Bangladesh may provide additional insight into the effectiveness of HEV 239 against these genotypes. As an RNA virus, HEV mutations are mainly associated with pathogenesis and susceptibility to antiviral drugs by enhancing HEV replication and infectivity. Few mutations have been demonstrated to be involved with viral immune escape [52]. On the other hand, the zoonotic potential of emerging genotypes highlights the importance of cross-genotype protection mediated by the hepatitis E vaccine. For instance, rat HEV is highly divergent from the subfamily *Orthohepevirinae*, sharing only 50–60% genomic identity [53]. Further study is needed in order to determine whether prior vaccine-induced immunity cross-protects against rat HEV.

## 9. Conclusions

Many efforts and trials have led to the ultimate successful development and clinical validation of the HEV vaccine. Expanding the coverage of HEV vaccination will be crucial in reducing the burden of HEV infections. To meet the global outbreak response strategy, issues related to WHO pre-qualification, age range expansion, the optimization of the immunization schedule under emergency, and vaccine transportation and administration remain to be addressed. Furthermore, additional clinical trials are required in order to evaluate the benefits and safety of the HEV vaccine in high-risk populations, particularly in pregnant women, individuals with underlying chronic liver disease, and children <16 years of age. Despite these issues, the successful deployment in South Sudan has demonstrated that contingency vaccination with the hepatitis E vaccine is practicable in complex emergencies, which may catalyze the use of the HEV vaccine in the future.

## Figures and Tables

**Table 1 vaccines-12-00719-t001:** Hepatitis E vaccines under clinical evaluation.

Vaccine	Manufacturer	Antigen	Express System	Dose	Efficacy	Development Status	Ref
HEV 239 (Hecolin)	Xiamen Innovax Biotech	HEV-1 ORF2 (aa 368–606)	*Escherichia coli*	0, 1, 6 m	100% (95% CI *, 72.1–100.0)	Licensed, Phase Ⅳ	[29,30,31]
rHEV	GlaxoSmithKline	HEV-1 ORF2 (aa 112–607)	*Baculovirus*	0, 1, 6 m	95.5% (95% CI *, 85.6–98.6)	Phase Ⅱ	[26]
Lipo-NE-P	Zydus Lifesciences Limited	HEV-1 ORF2 (aa 458–607)	*Escherichia coli*	0, 1, 6 m	-	Phase Ⅱ	[28]
HEV P179	Changchun institute of biological products Co., Ltd	HEV-4 ORF2 (aa 439–617)	*Escherichia coli*	0, 1, 6 m	-	Phase Ⅰb	[27]

* CI, 95% confidence interval.

## Data Availability

Data sharing is not applicable to this article.

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
