# Peer review of "Progress and Challenges to Hepatitis E Vaccine Development and Deployment"

_vaccines, 2024, doi:10.3390/vaccines12070719_

Round 1

Reviewer 1 Report

Comments and Suggestions for Authors

This article reviewed the progress and challenges for hepatitis E vaccines, especially for HEV 239 vaccine, the only licensed hepatitis E vaccine. The article was well-written. however, there are some issues should be addressed.

1. Page 2, Line 52, "foracute-on-chronic" should be "acute-on-chronic".

2. Reference[25] should be refined to provide more clear and specific information. This reference could also be displayed in Table 1. 

Comments on the Quality of English Language

Minor editing of English language required.

Author Response

1. Comments 1: Page 2, Line 52, "foracute-on-chronic" should be "acute-on-chronic".

    Response 1: Thank you for your comments. We have corrected the typo.

2. Comments 2:Reference[25] should be refined to provide more clear and specific information. This reference could also be displayed in Table 1. 

    Response 2: Thank you for your comments. We have refined the reference.

Reviewer 2 Report

Comments and Suggestions for Authors

The HEV vaccine is a powerful tool for the prevention of hepatitis E. In this review, Huang and colleagues have effectively summarized the development of the HEV vaccine and the challenges associated with it. Overall, this review is well-organized and well-written, providing crucial insights that could catalyze the future use of the HEV vaccine. I have only one minor point for this review:

Line 292: I suggest changing the heading to "8. Challenges in Hepatitis E Vaccine Development" instead of "Hepatitis Vaccines," which is too general.

Author Response

Comments 1: Line 292: I suggest changing the heading to "8. Challenges in Hepatitis E Vaccine Development" instead of "Hepatitis Vaccines," which is too general.

Response 1: Thanks for pointing this out, we've made the revision.

Reviewer 3 Report

Comments and Suggestions for Authors

Huang et al. interested in the development and advancement of hepatitis E vaccines, particularly the HEV239 (Hecolin) vaccine by a Chinese group. The authors summarized the progress of HEV239 in clinical trials and challenges of HEV239 encountered in special groups, such as pregnant women and patients with chronic liver diseases. Overall, HEV239 demonstrates remarkable efficacy and holds significant promise. Here are some minor suggestions for the authors' consideration:

1.        The major concern of this reviewer is that the manuscript actually primarily focuses on the HEV239. Consider revising the title and subheadings. A suggested title or similar: Progress and Challenges in Hepatitis E Vaccine Development and Deployment: A Focus on HEV239.

2.        The classification of HEV has already been updated. Please revise it accordingly (ICTV Virus Taxonomy Profile: Hepeviridae 2022).

3.        Rat HEV-C1 has also been identified in immunocompetent individuals (J. Infect. Dis. 2019;220:951–955).

4.        Some contents in Table 1 have not been described in the 2. Clinical Progress.

5.        Note that Tables have not been referenced in the main text.

6.        Have the authors checked Clinicaltrials.gov for information on hepatitis E vaccine trials?

7.        It is recommended to add a Figure of timeline of HEV239 development and clinical studies, which would be more readable than Tables.

8.        Standardize the nomenclature for genotype 1 HEV as HEV-1 and genotype 4 HEV as HEV-4 for consistency.

Author Response

Comments 1: The major concern of this reviewer is that the manuscript actually primarily focuses on the HEV239. Consider revising the title and subheadings. A suggested title or similar: Progress and Challenges in Hepatitis E Vaccine Development and Deployment: A Focus on HEV239.

Response 1: Thank you for your suggestion. This review summarizes the progress of clinical researches for HEV vaccines in development. We have tried to include almost all the published clinical data, however, the majority of data were from the comprehensive serial studies focus on HEV239, which is also the only licensed HEV vaccine. We will consider updating the review when more data for other vaccines are published.

Comments 2: The classification of HEV has already been updated. Please revise it accordingly (ICTV Virus Taxonomy Profile: Hepeviridae 2022).

Response 2: Thank you for your suggestion. We have updated the HEV classification information in the introduction.

Comments 3: Rat HEV-C1 has also been identified in immunocompetent individuals (J. Infect. Dis. 2019;220:951–955).

Response 3: Thank you for this comment. As you pointed out, a case study demonstrated that rat hepatitis E virus was associated with severe acute hepatitis in an immunocompetent patient. Nevertheless, more evidence indicated immunosuppressed populations are susceptible to persistent infection with rat HEV and extrahepatic manifestations (Clin Infect Dis. 2022;75(2):288-296). Consequently, we revised the wording of the susceptible population for rat hepatitis E.

Comments 4: Some contents in Table 1 have not been described in the 2. Clinical Progress.

Response 4: Thank you for your suggestion. We have updated the safety data from the Phase I study of the P179 vaccine (Changchun institute of biological prod-ucts Co.Ltd) into clinical progress. Another Lipo-NE-P vaccine, developed by Zydus Lifesciences Ltd, has not post results on ClinicalTrials.gov or in the academic literature.

Comments 5: Note that Tables have not been referenced in the main text.

Response 5: Thank you for your suggestion. We have checked.

Comments 6: Have the authors checked Clinicaltrials.gov for information on hepatitis E vaccine trials?

Response 6: Thank you for your suggestion. We've checked the information on hepatitis E clinical trials in Clinicaltrials.gov and included the relevant information in this review.

Comments 7: It is recommended to add a Figure of timeline of HEV239 development and clinical studies, which would be more readable than Tables.

Response 7: Thank you for your suggestion. Timelines provide a clear overview of the clinical progress of the HEV239 vaccine, but tables are more appropriate to summarize amounts of clinical trial data. We sorted the study start date in Table 2 chronologically and shifted them to the second column.

Comments 8: Standardize the nomenclature for genotype 1 HEV as HEV-1 and genotype 4 HEV as HEV-4 for consistency.

Response 8: Thank you for your suggestion. We have checked.

Reviewer 4 Report

Comments and Suggestions for Authors

In this review article, the authors have thoroughly described the development of the Hepatitis E virus vaccine. The article is well-written and well-summarized. I have a few suggestions to share with the authors.

  1. Now, it is widely known that HEV has eight genotypes. The authors can include some recent updates about genotype-7 and -8 in the introduction section. For example, HEV genotype-8 could experimentally infect cynomolgus macaques. Also, HEV genotype-1 has ORF4.
  2. The expanding knowledge of HEV genotypes and their host range highlights the importance of HEV vaccine-mediated cross-genotype protection. Against this background, HEV 239 still needs to be established about its cross-protection against newly evolving genotypes. These things must be discussed in the challenges section. 

Author Response

Comments 1: Now, it is widely known that HEV has eight genotypes. The authors can include some recent updates about genotype-7 and -8 in the introduction section. For example, HEV genotype-8 could experimentally infect cynomolgus macaques. Also, HEV genotype-1 has ORF4.

Response 1: Thank you for your suggestion. We have added the recent progress of HEV genotype-7 and -8 to the introduction section. Further descriptions of ORF-4 were also provided.

Comments 2: The expanding knowledge of HEV genotypes and their host range highlights the importance of HEV vaccine-mediated cross-genotype protection. Against this background, HEV 239 still needs to be established about its cross-protection against newly evolving genotypes. These things must be discussed in the challenges section. 

Response 2: Thank you for your valuable insight. Newly evolving genotypes infecting humans have been reported in recent years, and the cross-protective effect of hepatitis E vaccines against these subtypes requires further evaluation. We have added the discussion of cross-protection against the newly developed genotypes of HEV239 into the Challenges section.